



# First evidence of microplastics in Antarctic snow

Alex R. Aves[1,2], Laura E. Revell[1], Sally Gaw[1], Helena Ruffell[1], Alex Schuddeboom[1], Ngaire E. Wotherspoon[1], Michelle LaRue[2], and Adrian J. McDonald[1,2]

[1]School of Physical and Chemical Sciences, University of Canterbury, Christchurch, New Zealand
[2]Gateway Antarctica, School of Earth and Environment, University of Canterbury, Christchurch, New Zealand

**Correspondence:** Alex R. Aves (alexandra.aves@pg.canterbury.ac.nz)

**Abstract.** In recent years, airborne microplastics have been identified in a range of remote environments. However, data throughout the Southern Hemisphere, in particular Antarctica, are largely absent to date. We collected snow samples from 19 sites across the Ross Island region of Antarctica. Suspected microplastic particles were isolated and their composition confirmed using micro-Fourier transform infrared spectroscopy (µFTIR). We identified microplastics in all Antarctic snow samples

at an average concentration of 29 particles $L^{-1}$, with fibres the most common morphotype and polyethylene terephthalate (PET) the most common polymer. To investigate sources, backward air mass trajectories were run from the time of sampling. These indicate potential long-range transportation of up to 6000 kilometers, assuming a residence time of 6.5 days. Local sources were also identified as potential inputs into the environment, as the polymers identified were consistent with those used in clothing and equipment from nearby research stations. This study adds to the growing body of literature regarding microplastics as a

ubiquitous airborne pollutant, and establishes their presence in Antarctica.

## 1 Introduction

Over the last century plastics have become one of the most ubiquitous synthetic materials in the world, due to their versatility and durability. Despite their longevity, plastics degrade over time to produce microplastics (plastic particles <5 mm in diameter), and when present in the environment have the potential to cause significant ecological damage (MacLeod et al., 2021).

Microplastics have negative effects on marine organisms (Wright et al., 2013), and act as vectors for persistent organic pollutants and other toxic substances (Mato et al., 2001; Rios et al., 2007), which are harmful to marine environments and organisms (Hermabessiere et al., 2017). Microplastics have been recognised as widespread pollutants in the marine environment (Ryan, 2015) and are known to be damaging to terrestrial ecosystems, (de Souza Machado et al., 2018), while their small size and relatively low density also allows them to become airborne and transported over large distances (Evangeliou et al., 2020).

Airborne microplastics have been identified in atmospheric fallout in a range of urban (Dris et al., 2015, 2016; Cai et al., 2017; Klein and Fischer, 2019; Knobloch et al., 2021) and remote regions worldwide (Allen et al., 2019; Bergmann et al., 2019; Brahney et al., 2020). It is now understood that microplastics transition between marine environments, terrestrial environments and the atmosphere via the plastic cycle (Horton and Dixon, 2018; Brahney et al., 2021). This allows microplastics to reach locations far from anthropogenic sources, such as the Arctic (Bergmann et al., 2019), the Tibetan Plateau (Zhang et al., 2021),

European alpine regions (Allen et al., 2019; Bergmann et al., 2019; Materić et al., 2020, 2021) and conservation areas across



the continental United States (Brahney et al., 2020). Deposited microplastics may accelerate melting of the cryosphere when present on snow and ice in alpine or polar regions (Evangeliou et al., 2020). Microplastics may further influence climate by acting as cloud ice nuclei in the atmosphere (Ganguly and Ariya, 2019), and through their minor contribution to global radiative forcing (Revell et al., 2021).

Antarctica was largely untouched by humans until the early 20[th] century due to its inaccessibility, extreme environmental conditions and barriers such as the Antarctic Circumpolar Current (Tin et al., 2014; Gordon, 1971). While the human footprint has increased over the last century, Antarctica has been set aside as a place of peace and science and is thought of as the last remaining true wilderness on earth (Tin et al., 2016). Due to this, Antarctica can act as an indicator of physical, chemical, and biological effects caused from anthropogenic stresses (Huiskes et al., 2006). Research on microplastics in the Antarctic has

focused on the marine environment, where particles have been detected in deep sea sediments in the Weddell Sea (Van Cauwenberghe et al., 2013), marine sediments from the western Antarctic Peninsula (Reed et al., 2018) and the Ross Sea (Munari et al., 2017), south of the Polar Front (Cózar et al., 2014) and in the surface waters of the Southern Ocean and Antarctic Peninsula (Absher et al., 2019; Cincinelli et al., 2017; Isobe et al., 2017; Suaria et al., 2020; Waller et al., 2017; Lacerda et al., 2019). Microplastics were recently identified for the first time in a freshwater Antarctic Specially Protected Area (ASPA) on Livingston

Island, which is used for long-term ecological monitoring due to its pristine nature and use as a reference for inland water research (González-Pleiter et al., 2020).

To date there is little information available regarding the presence of airborne microplastics in Antarctica. We collected freshly fallen snow samples from the Ross Island region of Antarctica in late 2019, and analyzed them to quantify the presence and abundance of microplastics. Samples were collected close to two scientific research stations (Scott Base and McMurdo

Station), and from 13 field sites up to 20 km from the research stations. We identified polymer composition using μFTIR spectroscopy and analyzed air mass back-trajectories to identify the potential origins of sampled air masses. Further, we catalogued the composition of field equipment to understand local polymer sources.

## 2   Materials and methods

### 2.1   Field collection

Snow samples were collected in 500 mL stainless-steel bottles. Nineteen samples were collected in total with six from locations near research stations and thirteen from remote locations with minimal human disturbance (Fig. 1). Samples were collected using a stainless-steel scoop and funnel to fill each bottle with snow from the top 2 cm of the surface. Samples were stored in a -25 °C freezer at Scott Base and were kept chilled with dry ice during transit to New Zealand. Upon arrival in New Zealand, samples were stored at -20 °C.





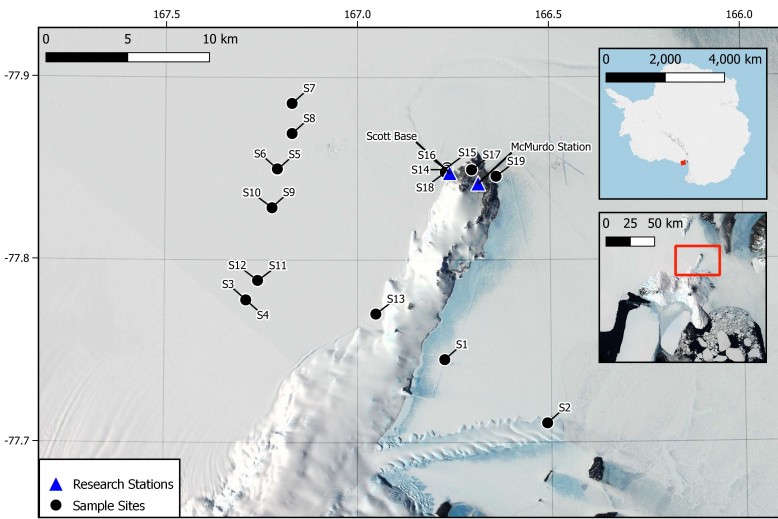

**Figure 1.** Site locations for snow sample collections across the Ross Island region of Antarctica (S1–S19). Sample sites are marked in black, and locations of scientific research stations shown by blue triangles. Some site markers correspond to two sampling sites due to the scale of the map. Map data sourced from Matsuoka et al. (2021).

## 2.2 Laboratory analysis of microplastics

Snow samples were thawed at room temperature for 24–48 hours prior to analysis. Thawed samples were filtered through a glass apparatus attached to a vacuum using a cellulose nitrate membrane filter (Whatman nitrocellulose membrane, 50 mm diameter, 0.45 µm pore size). Approximately 10–20 mL of 70% ethanol was used to rinse the filters, and a further 10–30 mL of 96% ethanol was added to soak the filter for 10 minutes, to prevent bacterial and viral growth for further biosecurity measures. The sides of the glassware were thoroughly rinsed with ultra-pure water (<18 MΩ) to dislodge any microplastics adhered to the walls of the filtering equipment, and samples were dried under vacuum.

The dried cellulose nitrate filter was transferred to a 250 mL glass beaker for a wet peroxide oxidation (WPO) digestion to remove organic material present on the filter. An Iron (Fe(II)) sulfate solution (0.05 M) was prepared by adding 7.5 g of $FeSO_4 \cdot 7H_2O$ (= 278.02 g mol$^{-1}$) to 500 mL of ultra-pure water and 3 mL of concentrated sulfuric acid (Fenton's reagent). This study followed similar methods to those outlined by the National Oceanic and Atmospheric Administration (NOAA) (Masura et al., 2015) for WPO digestion, with 20 mL of 30% $H_2O_2$, 100 mL ultra-pure water and 5 mL of the Fe(II) solution added to the beaker with the filter. The solution was then left, with the beaker opening covered in aluminium foil, for 20 minutes before being stirred on a hot plate at 45 °C for 2–3 hours with a magnetic stir bar.



The digested cellulose nitrate filter was rinsed with ultra-pure water to remove any attached material. The digested filtrate was

then filtered under vacuum onto a Whatman GF/C filter (47 mm diameter, 1.2 µm pore size) and all glassware was thoroughly

rinsed with ultra-pure water. The GF/C filter was then vacuum dried before being removed with tweezers into a closed petri

dish and labelled accordingly for storage until analysis.

## 2.3    Visual characterisation of microplastics and µFTIR analysis

Filter papers were initially screened using a Leica MZ125 stereomicroscope with $10\times$ magnification for visual identification.

Suspected plastic particles were identified according to characteristics related to the morphotype, shape, colour and the physical

characteristics of each particle (Bridson et al., 2020). Microplastics were characterized into four main morphotypes – fibres,

films, fragments and beads. Colours were recorded for each suspected particle. Analysis was performed by the same individual

to ensure consistency in identification and counts. Each filter was visually analyzed three times. Due to limitations of visual

identification techniques (Knobloch et al., 2021), dark colours were difficult to differentiate. Therefore throughout this study,

'blue' includes blue, black and navy.

All suspected microplastics were chemically identified by micro-Fourier transform infrared spectroscopy (µFTIR). This

procedure followed pre-existing methods for characterizing microplastics (Primpke et al., 2018). A Hyperion 2000 µFTIR

microscope (Bruker Optics), attached to a Vertex 70 (Bruker Optics) spectrometer, was used to analyse particles plated on a

calcium fluoride ($CaF_2$) disk (25 mm diameter), with a liquid nitrogen cooled Mercury-Cadmium-Telluride (MCT) detector.

Each particle was manually transferred onto the disk using tweezers and a drop of 96% ethanol to aid in transfer. An opti-

cal overview image was recorded before infra-red measurements were performed at $15\times$ magnification. Scans were run in

transmission mode (10 scans, 4 cm$^{-1}$ resolution, spectral range of 4000–1000 cm$^{-1}$) using OPUS 7.8 software. All spectra

were saved and run against a Wiley spectral library (Databases: HIX, OSX, HDX, PPX, KLX, FMX, YX, DAX, HUX, QPX,

WSAI1X). Particles returning a hit quality index (HQI) of >70% against the library reference spectra were accepted as mi-

croplastics. Those <70% that exhibited plastic characteristics from visual screening and similar µFTIR spectra were analyzed

further using Wiley peak picking tools to identify characteristic peaks of plastic polymer types.

## 2.4    Quality control

### 2.4.1    Field sampling

Each sampling site was selected ensuring the presence of fresh snow with no visible contamination or movement in the col-

lection area. Non-plastic sampling equipment was used to avoid contamination and all equipment was rinsed in nearby snow

prior to each sample collection. Leather gloves were worn by the sample collectors and the sample site selected upwind from

any human movement to minimise contamination by the samplers. The lid of each sampling bottle was held without the inside

being touched during sample collection to avoid human contamination. Bottles were stored upright and kept in a cool box in

snow during transportation.



### 2.4.2   Controls and blanks

Two field controls were collected during sample collection: one alongside a remote field site sample and one alongside a research station sample. The two field control bottles were left open during snow sampling and filled with ultra-pure water when returned to the laboratory. They subsequently underwent the same laboratory methods as all other samples. Two method controls were prepared in stainless-steel bottles identical to the sampling bottles used for collection and were filled with ultra-pure water. These were stored in the laboratory freezer in New Zealand during sampling and underwent identical laboratory procedures as the samples to identify potential sources of contamination from the sampling bottles.

### 2.4.3   Laboratory analysis

Method recovery tests consisted of two 500 mL samples of ultra-pure water spiked with 5 polyethylene (PE) beads and 7 polymethyl methacrylate (PMMA) fibres sized between 500–2000 μm. The spiked samples were analyzed identically to the field samples to measure recovery rates of the spiked samples. Daily laboratory blanks were analyzed using the same procedure as all samples with 500 mL of ultra-pure water. Filtration and laboratory procedures were performed in a Labrocare fume hood cabinet to limit contamination by airborne microplastics. Glassware was cleaned inside the fume hood three times with ultra-pure water and once with acetone. Aluminium foil was used to cover glassware openings to minimise contamination by airborne microplastics. The laboratory blanks were analyzed to compare results for each individual date and accounted for in the data for the samples filtered on the corresponding days. Non-synthetic (wool and cotton) clothing was worn during the laboratory and analysis process. Particles found in field samples with identical characteristics to those found in blanks were discarded and excluded from the results. Surfaces were sprayed and wiped down with 70% ethanol prior to microscopy and analysis work, with filters remaining covered except when manual extraction of particles occurred.

### 2.5   Clothing composition

National Antarctic programs provide essential clothing and field gear for staff and scientists. Some of the gear provided is mandatory whenever undertaking fieldwork or travelling outside of bases. The composition of field gear provided by the New Zealand National Antarctic program (including base layers, mid layers, outer layers, shoes, boot liners, gloves, bags, hats and accessories) was catalogued to determine potential local sources of synthetic particles into the Ross Island region (Table A1).

### 2.6   Trajectory analysis

Because samples were collected during, or shortly after a single snowfall event (Fig. 2), air mass trajectories were produced to understand potential source regions. Lagrangian air parcel trajectories were derived using the Hybrid Single Particle Lagrangian Integrated Trajectory Model (HYSPLIT) (Stein et al., 2015). HYSPLIT was run using meteorological data from the National Oceanic and Atmospheric Administration Global Forecast System (GFS), with a horizontal grid resolution of $0.25° \times 0.25°$ and a temporal resolution of 3 hours. To achieve this temporal resolution the GFS model is initialized from observational data every 6 hours and then the three hour forecast is included in the GFS output to fill the gaps in time (Stein et al., 2015).

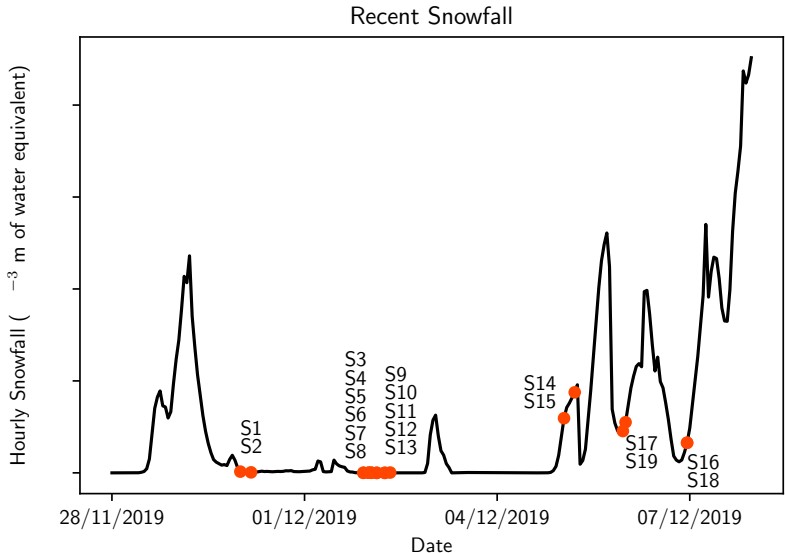

**Figure 2.** Snowfall over the measurement period from the ERA5 reanalysis. These values are averaged over the sample site area (77.75–78°S, 166.5–167.25°E). Orange circles indicate the time when samples were taken with nearby text indicating the sample number.

The ensemble configuration of HYSPLIT was used to generate 27-member back-trajectory ensembles, which represent the uncertainty in the trajectories. For each ensemble trajectory, the meteorological data was offset in the x and/or y directions by one grid point, and/or 0.1 sigma units in the z direction, with 27 ensemble members covering all possible combinations of these offsets. Back trajectories were run from starting points at each of the sampling sites at the corresponding time of sampling and

with starting heights of 500, 1000 and 2000 m. Recent published results show that the residence time for microplastic particles in the atmosphere may vary between 1 hour and 156 hours (Brahney et al., 2021). As such, the back trajectories were run for 156 hours to show the full range of possible sources.

## 3 Results

### 3.1 Blanks and sample extractions

Recovery rates for spiked samples were 100%. Across the sample controls an average of one particle was found in daily laboratory blanks ($n$=11), three particles in field blanks ($n$=2) and two particles in method controls ($n$=2). In all blanks, fibres were the most common morphotypes. Several suspected microplastic fragments present in the controls and samples resembled that of the sampling vessel. Spectroscopic analysis confirmed that the outside bottle coating and fragments found in the blanks were polymethyl methacrylate (PMMA). Suspected microplastic particles in field samples of an identical colour and morphotype to

those detected in the field blanks were not analyzed further and discounted from the results. All reported PMMA still included

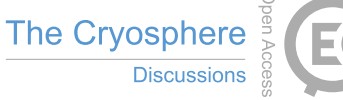



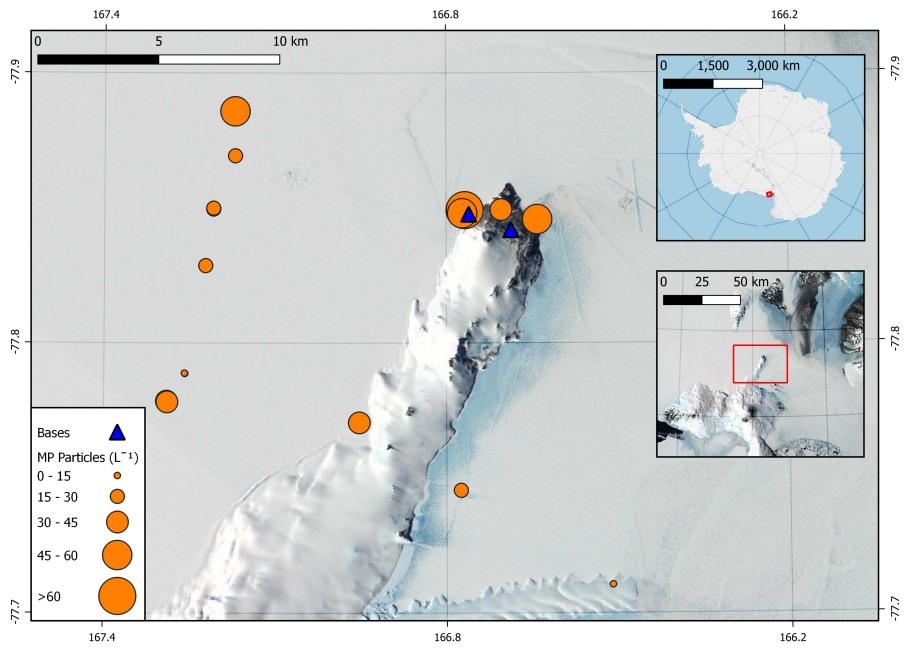

**Figure 3.** Concentrations of microplastics (MP) particles $L^{-1}$ at each sampling site in the Ross Island region. Insets provide an overview of the location of sampling sites in Antarctica. Map data sourced from Matsuoka et al. (2021).

in results was a different morphotype and colour to the corresponding sampling bottles, and therefore some PMMA is still shown in results.

### 3.2 Concentration of microplastics in snow samples

Microplastics were found in all Antarctic snow samples with a total of 109 particles confirmed as microplastics using µFTIR
spectroscopy across the 19 field samples (Fig. 1). Microplastics were present at an average concentration of 29.4±4.7 particles $L^{-1}$ of melted snow (mean ± one standard error) across all sites (Fig. 3). The average microplastic concentration was 22.5±4.0 particles $L^{-1}$ and 47.2±8.4 particles $L^{-1}$ at remote sites and base sites, respectively (Fig. 3, Supplementary Fig. 1). The highest concentration found was 82 particles $L^{-1}$ at site S16 (Scott Base) and the lowest concentration found was 4 particles $L^{-1}$ at site S2 (Erebus Glacier Tongue).

### 3.3 Characterization of microplastics

A total of 13 different polymer types were identified across the snow samples when compared against a spectral reference library (Supplementary Fig. 2). Polyethylene terephthalate (PET) was the most frequently detected polymer type, found in 79% of the samples, comprising 41% of total polymers identified (Fig. 4a). Copolymers (CP, those containing two or more different monomers) equated to 17% of total polymers identified (Fig. 4a and Table A2). Polymer types comprising less than 10% of

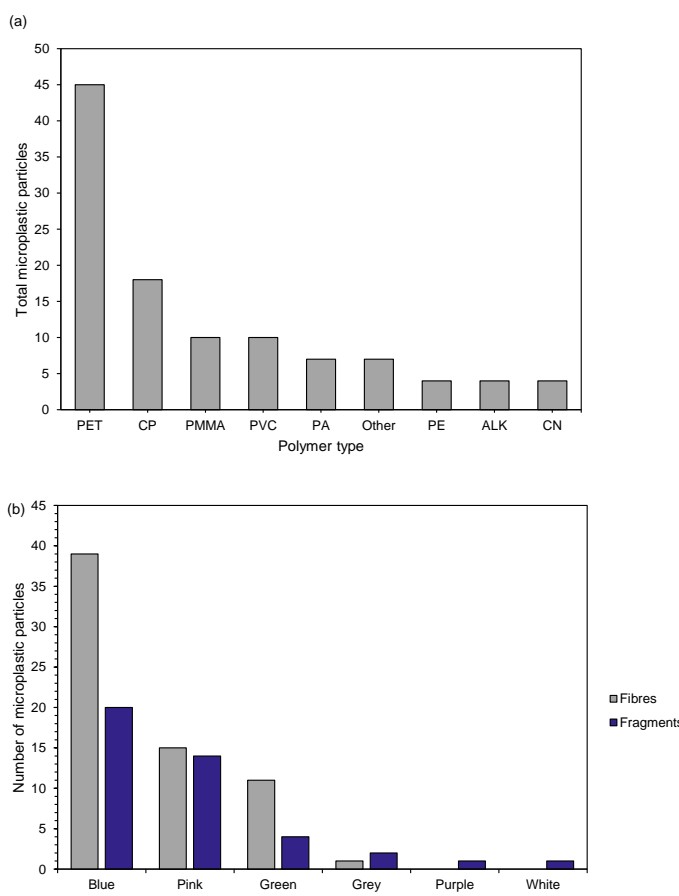

**Figure 4.** Polymer types identified across all samples. (a) Number of each polymer type found across all Antarctic study sites. (b) Number of microplastic fragments and fibres identified in each colour category (films (*n*=1) excluded).

the total polymers identified included polymethyl methacrylate (PMMA, 9%), polyvinyl chloride (PVC, 9%), polyamide (PA, 6%), polyethylene (PE, 4%), alkyd (ALK, 4%), cellulose nitrate (CN, 4%) and other (6%). Polymer types present in the 'other' category with a detection frequency less than 4% include polytetrafluoroethylene (PTFE), polyvinylidene, polypropylene, silicone and polymethyl anhydride. The cellulose nitrate particles detected were not found in laboratory control samples and did not match the morphotype and colour of the filters used in the digestion stage, so were included in results.


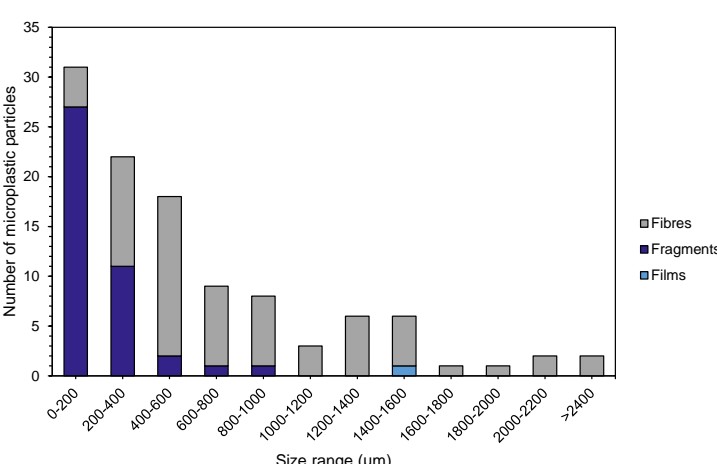

**Figure 5.** Size distribution of microplastics across all samples categorized by morphotype (length).

Particles confirmed spectroscopically as microplastics were classified as fibres, fragments, and films (Supplementary Fig. 3). No beads were detected. Fibres were the most abundant morphotype (61%, Fig. 5) and the majority of microplastics identified were blue (55%) and pink (23%, Fig. 4b). Blue was the most common colour of both fibres and fragments. The size range of microplastics detected varied between 50–3510 μm with an overall average size of 606 μm (Fig. 5). A large proportion of microplastics were <1000 μm (81%), with 28% in the 0–200 μm size range (Fig. 5). All fragments were ≤1000 μm while fibres





were present in all size groups (Fig. 5). The mean size for fragments was 200 μm while the mean size for fibres was 850 μm, which skews the total averages (Fig. 5).

### 3.4 Snowfall and back trajectories

All of the samples from remote sites (S1–S13) were collected over three days (30 November – 2 December 2019). A large snowfall event occurred one day prior to sampling on 30 November 2019 (Fig. 2), and no snowfall was reported over the

subsequent two days, therefore all remote samples originated from the same snowfall event. Because only the top 2 cm of snow was collected, the microplastics we found were likely deposited in the recent snowfall event.

Figure 6 shows Lagrangian back-trajectories beginning at the time and location of six of the different sampling sites. The six events shown were chosen to cover the largest spatial distances and time periods within the data set, with many of the excluded sites showing near-identical results to included sites. This figure only shows the results for trajectories that start with an altitude

of 2000 m, but trajectories initialised at 500 m and 1000 m show very similar results (Fig. A1 and A2). Distances travelled by air masses over 6 hours, 24 hours and 156 hours are shown in Table A3 to represent a range of potential airborne microplastic residence times.

The trajectories presented in Fig. 6 indicate that the snowfall event was likely linked to a Ross Air Stream event, where strong near-surface winds flow from the southeast parallel to the Trans-Antarctic Mountains (Parish et al., 2006). The Ross Air

Stream is primarily fed by air from the Siple Coast region in West Antarctica, caused by a complex mixture of katabatic winds and barrier flows along the Trans-Antarctic mountains created by cyclones (Seefeldt and Cassano, 2012). Pressure gradients induced by cyclones to the north of the ice shelf force air to flow towards the Trans-Antarctic mountains, but the resulting wind lacks the kinetic energy to pass over the mountains causing a barrier flow. These conditions occur approximately 24% of the time, though are more common in the Austral winter, and bring warmer temperatures and strong winds to the Ross Island

region (Coggins et al., 2014) along with clouds connected to snowfall (Jolly et al., 2018). The splitting of the Ross Air Stream around Ross Island also impacts the formation of the Ross Sea polynya and McMurdo Sound polynya (Dale et al., 2017; Brett et al., 2020). These open water regions provide a possible secondary local source of microplastics; in particular, the McMurdo Sound polynya is very close to site S2. The back trajectories shown in Fig. A1 for Sites 3, 7 and 13 could mean that the open water in the Ross Sea polynya is a relevant source.

In accordance with the climatology of the region, all of the trajectories show that the immediate source of the airflow is from the south following the Trans-Antarctic mountains. This means that for the majority of these samples, short term local transport is the most likely source of microplastics as the sampling sites are mostly north of the local bases (Scott Base and McMurdo Base). It is also possible that local small scale transport processes that are not captured by HYSPLIT could play a key role in transport at this scale. For the trajectories that clearly show a Ross Air Stream event (sites 2, 3, 7 and 13), the most likely distant

sources are the Ross and Amundsen Seas as there are no manned stations or other likely sources along the trajectory path. The transport processes in this region can be very rapid with the trajectories from these sites on average covering a distance of 143 km in the first 6 hours and 469 km in 24 hours. With a 156 hour residence time, transport over thousands of kilometers is possible.

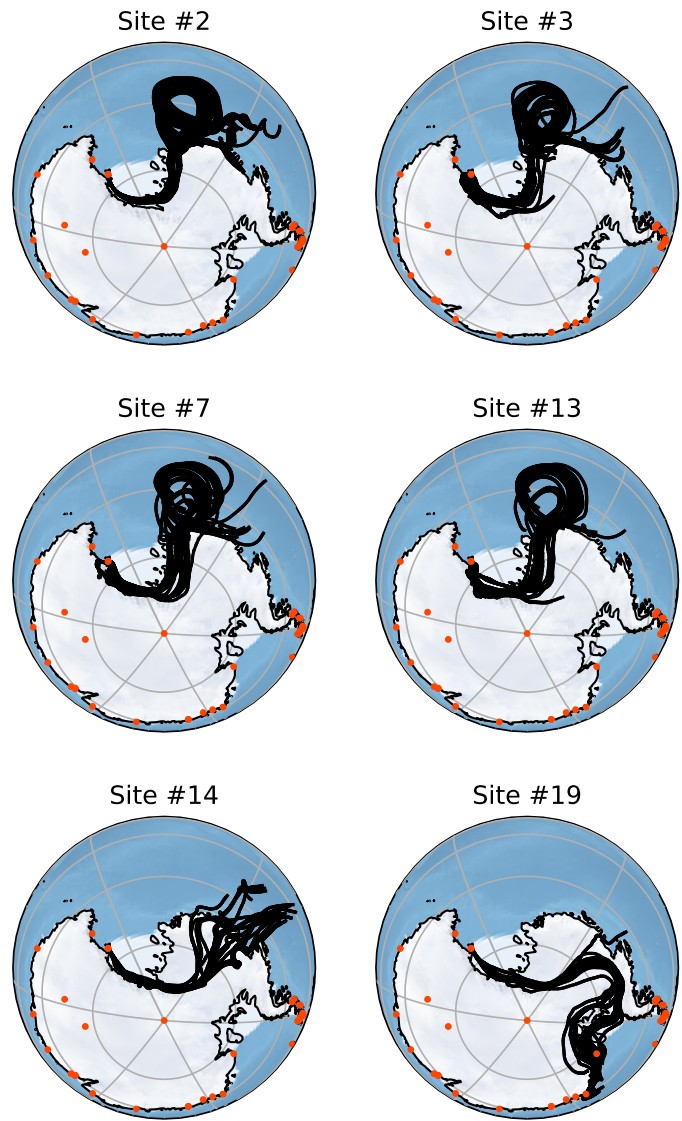

**Figure 6.** Back-trajectory ensembles generated by HYSPLIT for six of the collected samples. Each of these trajectories is run for 156 hours with the ensemble members generated by minor perturbations as described in section 2.6. Orange points indicate the locations of active Antarctic research stations that operate all year. All of these trajectories are started at 2000 m altitude but show very similar results to those started at 500 m and 1000 m. Note that each simulation is run to correspond to the time of sampling so is host to different meteorological conditions than other simulations.





For site 19 there is a different set of meteorological conditions causing a divergence from the results from the other samples.
The corresponding trajectories show a wide range of possible sources including the Antarctic Peninsula and Weddell Sea.
While these conditions are relatively rare compared with the kind of Ross Air Stream flow examined earlier, they still provide a
possible source of microplastics. These different conditions expand the set of possible sources to include more distant sources
including many other Antarctic research stations. Given that an assumed maximum residence time of 156 hours leads to
trajectories covering a total distance of over ∼4800 km in transit (Table A3), transport from these stations appears possible but
unlikely.

## 4 Discussion

### 4.1 Microplastic concentrations

Our work provides the first evidence of microplastics in Antarctic snow, and critically, the average concentration of microplastics found in this study are higher ($29.4\pm4.7$ particles $L^{-1}$) than in the surrounding Ross Sea ($1.7\times10^{-4}$ particles $L^{-1}$) and
those reported in East Antarctic sea ice (11.7 particles $L^{-1}$) (Kelly et al., 2020; Cincinelli et al., 2017). Our reported concentrations of microplastics in surface snow near two Antarctic stations ($47.2\pm8.4$ particles $L^{-1}$; highest concentration 82.1 particles $L^{-1}$) identified similar concentrations to those present in supraglacial debris from an Italian glacier (74 particles $kg^{-1}$ dry weight) (Ambrosini et al., 2019). For comparison, concentrations of microplastics in Arctic sea ice have been shown to range from 8 to 41 particles $L^{-1}$ (Geilfus et al., 2019), while Arctic snow sampling identified higher concentrations of mi-
croplastics up to $14.4\times10^{3}$ particles $L^{-1}$ (Bergmann et al., 2019), which may be attributed to the proximity to more populated regions.

### 4.2 Microplastic characteristics

The most frequent polymer type we detected was PET, which was found in 79% of all samples. Approximately 60% of all PET produced are used for synthetic fibres and 30% for plastic bottles (Ji, 2013). Our results differ to surrounding marine studies
which found PE and PP were the most common polymer types in seawater samples from the Ross Sea region (Cincinelli et al., 2017). Comparatively, varnish (including acrylates) and rubber were the most common in Arctic snow collected using similar sampling techniques, with PMMA being the third most common polymer type in this study (Bergmann et al., 2019). Air samples collected over the West Pacific Ocean similarly found PET was present in the highest abundance (57%) (Liu et al., 2019b).
Fibres were the most abundant morphotype (60%) followed by fragments (39%) and films (1%, Fig. 5). These findings are consistent with previous studies whereby fibres were also the dominant morphotype of airborne microplastics due to their relatively low density and physical characteristics (Liu et al., 2019a; Dris et al., 2016; Bullard et al., 2021). Analysis of seawater samples in the surrounding Ross Sea identified fragments as the predominant morphotype (72%) (Cincinelli et al., 2017), in contrast to our results. Fibres were the most predominant morphotype in sediment from Terra Nova Bay (Munari et al., 2017),





which is consistent with the findings of this research. This likely indicates that the distribution of microplastics around the Ross Sea region is heterogeneous depending on the sample type. Further research is required to understand the microplastic footprint in the region. Similar to previous studies, we identified darker colours (blue, black and navy) as the most common (Ambrosini et al., 2019; Liu et al., 2019a, b). Dark-coloured microplastics are likely efficient at absorbing solar radiation compared to lighter colours and are of particular concern in the cryosphere as they may accelerate melting (Evangeliou et al., 2020).

As the size distribution of identified microplastics is skewed towards smaller particles (Fig. 5), it is likely that particles smaller than the smallest particle observed (50 µm) are present, but not able to be detected due to the magnification limit of the stereomicroscope (20 µm) and difficulties in handling particles <50 µm. The abundance of microplastics has previously been shown to increase with decreasing size (Isobe et al., 2017; Levermore et al., 2020), which corresponds to the findings of this study (Fig. 5). The size distribution from our study was comparable to those measured in the remote Pyrenees (Allen et al.,

2019). We identified that only 10% of the microplastics were <100 µm while findings in the Arctic reported 98% of particles were <100 µm (Bergmann et al., 2019). While fibres were present across all size ranges found, the only fragments found were smaller than 1000 µm, consistent with previous studies (Revell et al. (2021) and references therein).

### 4.3 Origin of microplastics

Microplastics in Antarctica may originate from both local sources and long-range transport. Direct sources of microplastics
to the Antarctic environment may include fragmentation of plastic equipment from research stations, clothing worn by base staff and researchers, and mismanaged waste. Microplastics may also enter the Antarctic environment via long range transport by ocean currents (Fraser et al., 2018), ocean to atmosphere exchange (Allen et al., 2020) and both short and long-range atmospheric transportation (Evangeliou et al., 2020; Brahney et al., 2021).

#### 4.3.1 Local sources of microplastics

Antarctic research stations on Ross Island, Scott Base (NZ) and McMurdo Station (US), are within the closest proximity to the sampling sites (Fig. 1), with Zucchelli Station (Italian) the next closest at 350 km away, which is only operational over summer. Direct transport from Zucchelli station is highly unlikely given that it is north from our sampling sites and the winds primarily come from the south. McMurdo Station has a maximum capacity of 1,300 people which is met in the summer months, and decreases to approximately 300 people over winter, whereas Scott Base has a capacity of 86 people over summer and typically
hosts 12 staff over winter. The concentration of microplastics measured at the base sites (S14-19; 47 $\pm$8 particles L$^{-1}$) was higher on average than the remote sites (22 $\pm$4 particles L$^{-1}$) suggesting that microplastics originated from local sources.

Plastic products in use at research stations (including building materials, marker flags, safety equipment and tyre rubber) may fragment with environmental exposure and be a potential local source of microplastics into the environment. General wear and weathering from clothing and outdoor equipment used in the field may introduce plastics into more remote regions away from
populated bases. In addition, enhanced ultra-violet fluxes due to the Antarctic ozone hole may accelerate the fragmentation of larger plastic products into microplastics (Williamson et al., 2019).





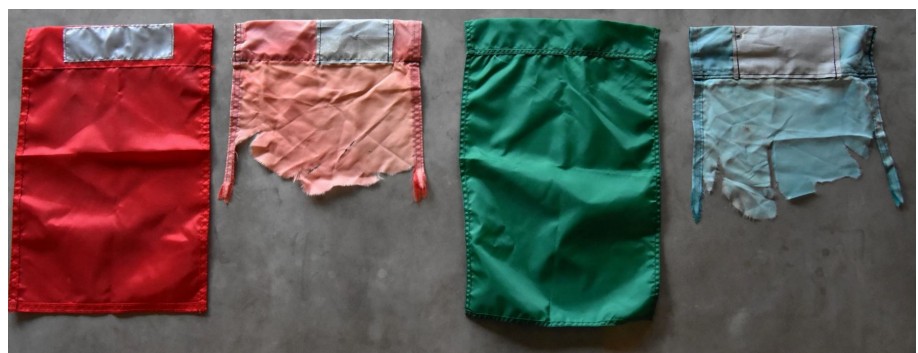

**Figure 7.** Marker flags used in Antarctica at McMurdo Station. Before being placed outside (left) and after they were retrieved (right). Photo credit: Evan Thompson.

PET was the most common polymer found in snow samples making up 41% of total microplastics. The most common polymer found in catalogued gear was also PET which was present in 48% of the garments in varying percentages from 68-100% followed by PA which was present in 30% of the garments (Table A1). Fibres were also the most common morphotype

in snow samples (Fig. 5) suggesting an origin from textile garments (Acharya et al., 2021). The most common colours used in the clothing are black, orange, navy and grey (Table A2). These colours are consistent with those found in snow samples, with 55% of the particles being of a dark colour (black, blue, navy; Fig. 4b). Throughout some regions of Antarctica, polyamide flags are used to identify safe routes for travel and are visible in harsh weather conditions. Weathering of wayfinding flags may release microplastics into the environment, and due to the high volume used each year could have the potential to impact

the surrounding environment. The flags are predominantly red, green, blue and black, with weathering processes resulting in a dulling of the flag colour to a pale pink or blue (Fig. 7). In this study, blue was the most common colour detected (55%) followed by pink (23%, Fig. 4b). Replacing the polyamide fabric used for wayfinding flags with an alternative non-synthetic material could reduce the local impact of microplastics on the environment caused by research stations in Antarctica.

Wastewater treatment plants (WWTPs) have been identified as a source of entry of microplastics to the environment world-

wide (Browne et al., 2011). An excess of 700,000 synthetic fibres are released from an average 6 kg load of washing acrylic fibres (Napper and Thompson, 2016). The WWTP at Zucchelli Station has been hypothesised as a potential source of microplastic fibres into the Ross Sea while identifying microplastics in nearby benthic sediment (Cincinelli et al., 2017), however no WWTPs in Antarctica have been investigated for their contribution of microplastics to the environment to date. The Pro-





tocol on Environmental Protection to the Antarctic Treaty (Annex III and Annex IV) entered into force in 1998. The Protocol
prohibits plastic waste from being released by ocean vessels within the Antarctic Treaty area and untreated sewage within 12
nautical miles of land or ice shelves. However, no annex for the release of effluent from research stations was established.
WWTPs are not mandatory at stations in Antarctica, leaving many research stations with insufficient wastewater treatment
facilities (Gröndahl et al., 2009; Stark et al., 2016). The findings of the Antarctic Treaty Consultative Meeting (ATCM) in 2019
recommended that all governments with Antarctic interests eliminate products containing microplastic beads, work towards
reducing microplastic release from wastewater systems and support monitoring of plastic pollution regarding human activity.

### 4.3.2   Long-range transport of microplastics

The transportation of particulates, such as dust, across the Southern Ocean and into Antarctica from other continents has been
explored in previous literature, identifying mid-latitude circumpolar westerly winds dominating atmospheric transport to these
regions, with Patagonia and New Zealand the most likely sources (Neff and Bertler, 2015). Transportation of microplastics
into the Southern Ocean via deep water processes can also introduce synthetic pollutants from afar (Mountford and Maqueda,
2020). Atmospheric long-range transport has been identified as a contamination source of persistent organic pollutants into
Antarctica (Kallenborn et al., 2013) as well as dust originating from Australia, Patagonia and the Northern Hemisphere (Li
et al., 2008).

Short-range transport of microplastics from the bases to the sampling sites is more likely than long-range transport, given
the sites' proximity to research bases and the climatology of the area. HYSPLIT trajectory modelling indicates that micro-
plastics may have travelled from the Amundsen or Ross Seas to reach the remote sample sites and could possibly have come
as far as the Weddell Sea (based on hypothesised particle residence time in Brahney et al. (2021)). There are no significant
anthropogenic sources identified along the trajectories of the majority of the air masses, such as other research stations, with
the Ross Ice Shelf extending for approximately 800 km westward from the sampling sites.

Microplastic particles may have originated from local anthropogenic sources and been transported around Antarctica via a
cycle of entrainment and deposition. Alternatively, microplastics may have originated from surface waters surrounding Antarc-
tica via co-emission with sea spray (Allen et al., 2020) and atmospheric transport. Given the air mass trajectories pass through
the Ross Sea, Amundsen Sea and potentially the Weddell Sea, these are all possible sources with known presences of plastics
and microplastics at these locations (Cincinelli et al., 2017; Munari et al., 2017; Van Cauwenberghe et al., 2013; Barnes et al.,
2010). Recent atmospheric transport modelling indicates that Antarctica is a net importer of microplastics, with the flux of mi-
croplastics from mismanaged plastic waste in the ocean transferring to the atmosphere at the Antarctic coast likely exceeding
anthropogenic sources of microplastics on the continent (Brahney et al., 2021).

### 4.4   Implications and outlook

The implications of microplastics reaching remote regions such as Antarctica are vast. Antarctic organisms have adapted to
extreme environmental conditions over many millions of years (Peck, 2018) and the rapid environmental changes due to an-
thropogenic influence is threatening the unique ecosystems present in the polar regions (Aronson et al., 2011; Convey and



Peck, 2019; Lee et al., 2017). Organisms' exposure to microplastics can lead to limited growth, negative effects on repro-
duction and impaired biological functions (Foley et al., 2018), increasing pressure on ecosystems. Epiplastic communities of
bacteria, microalgae and invertebrates can form on environmental plastic particles, which may contribute to the movement and
introduction of invasive species into the Antarctic region (Lacerda et al., 2019, 2020).

Negative effects of plastic pollution in Antarctic waters have been reported since 1990 when anthropogenic products, mainly
from fishing vessels, were found to be entangling fur seals (Croxall et al., 1990). The ingestion of microplastics by zooplankton,
an ecologically important organism in marine ecosystems, can disrupt usual biological processes and negatively impact upon
function and health (Cole et al., 2013). The ingestion of microplastics by Antarctic krill may cause a multitude of negative
effects on the entire Antarctic food chain as they are a keystone species which a large number of organisms rely on to survive
in the Southern Ocean (Hill et al., 2006). Plastic ingestion by the common Antarctic collembolan (*Cryptopygus antarcticus*)
has also been identified, indicating the presence of plastic in the Antarctic terrestrial food chain (Bergami et al., 2020).

The consumption of microplastics by higher predators in the Antarctic ecosystem has also been noted, with gentoo (*Py-
goscelis papua*), Adélie (*Pygoscelis adeliae*), chinstrap (*Pygoscelis antarcticus*) and king (*Aptenodytes patagonicus*) penguins
showing the presence of microplastics in their diet (Fragão et al., 2021; Le Guen et al., 2020). Atmospheric transport of mi-
croplastics increases availability to terrestrial organisms, with the potential for ingestion and consequently, negative health
effects. Endemic Antarctic penguin species could be exposed to microplastics during their breeding season which is spent on
Antarctic fast ice with their colonies. The compounding effects of anthropogenic changes are putting Emperor penguin species
at risk with current models predicting a population decline of 81% by 2100 (Jenouvrier et al., 2019).

Microplastics are an emerging contaminant for which appropriate controls and regulations have not been widely put in
place (Waller et al., 2017). Members of the Antarctic Treaty committed to reducing plastic pollution in Antarctica and the
Southern Ocean at the Antarctic Treaty Consultative Meeting in 2019, urging Antarctic activities to: reduce their use of plastic
care products containing microplastics, reduce release of microplastics from WWTP and support greater monitoring of plastic
pollution in the region. As microplastics continue to pose a growing threat to the Antarctic ecosystem more rapid approaches
are required throughout the Antarctic Treaty System to minimise the widespread impacts. Further research is required to
determine the inputs of microplastics into the Antarctic environment, as well as developing a greater understanding of the role
of long-range transport for microplastic distribution.

The findings of this study highlight the global reach of plastic pollution and identifies the need for urgency in creating
successful policy to reduce its extent and effects, both globally and locally. While studies in the Antarctic region currently
focus on marine microplastic pollution, future research and policy needs to take a holistic approach to incorporate airborne and
terrestrial impacts. The Protocol on Environmental Protection to the Antarctic Treaty (1991) aims to promote the protection
of the Antarctic environment and its place in the world as a natural reserve devoted to peace and science. Growing rates of
pollution across the world make this a much greater challenge, and huge transdisciplinary efforts are required to ensure this
can continue to be achieved.



## 5 Conclusions

This study confirms the presence of microplastics in Antarctic snow. Microplastics were identified on the Ross Ice Shelf and near Scott Base and McMurdo Station at an average concentration of 29 particles $L^{-1}$. Fibres were the most common morphotype identified, and PET was the most common polymer found. Outdoor clothing and other equipment used at the nearby research stations was catalogued to understand local inputs of microplastics, while also assessing the potential for long-range transport through back-trajectory modelling. Back-trajectories indicated that microplastics may have travelled a distance of up to 6000 km, depending on the residence time. Given that air masses passed over the bases prior to sampling, and that the polymers we identified are consistent with those catalogued, it is likely that the majority of identified microplastics originated from local inputs from surrounding research stations. Our results highlight the importance for further monitoring – both in Antarctica and in surrounding waters – to develop our understanding of the microplastic footprint in Antarctica and the threat they may pose to the Antarctic environment.



# 6 Appendix

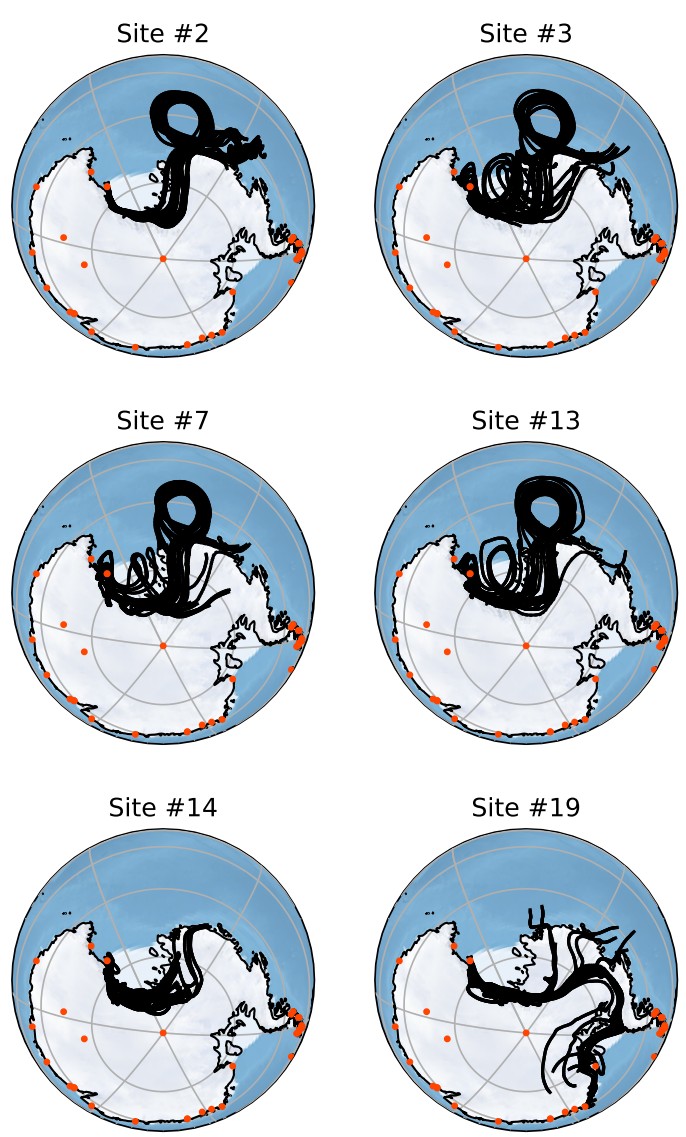

**Figure A1.** As for Fig. 6, but trajectories were initialised with a starting height of 500 m.


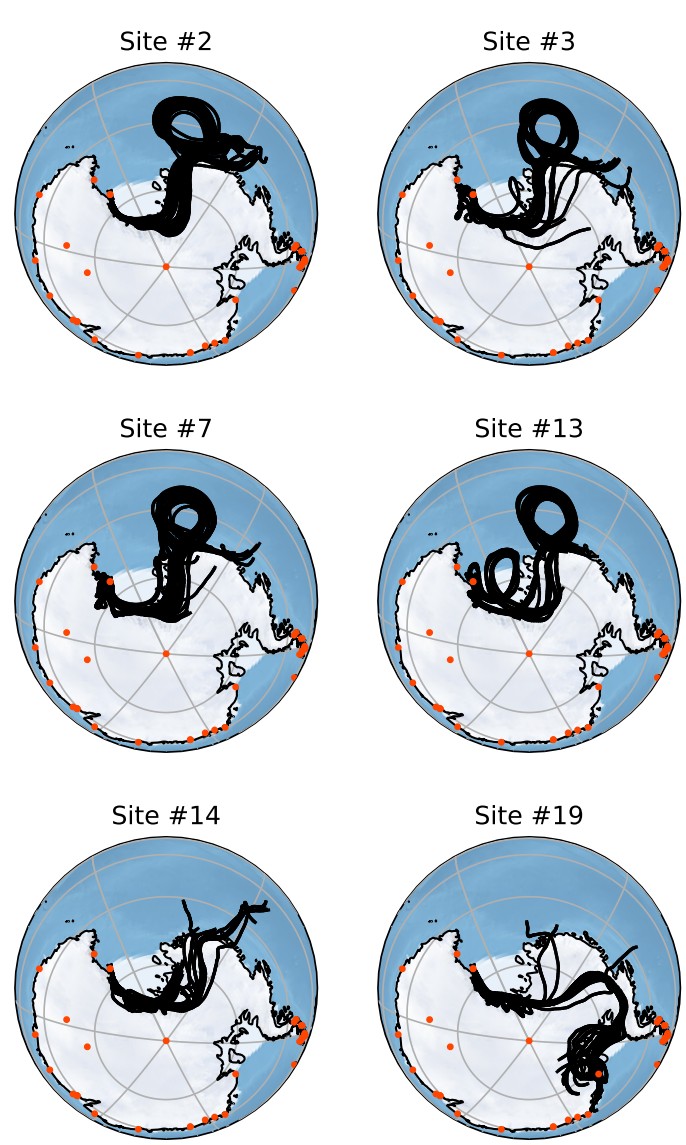

**Figure A2.** As for Fig. 6, but trajectories were initialised with a starting height of 1000 m.





| Garment | Material composition | Notes |
|---|---|---|
| Base layers (tops and bottoms) | 100% merino wool | Elasticated waste band and stitching likely to be nylon |
| Fleece shirt (full zip) | 100% polyester | Colour: black |
| Fleece pants | 100% polyester | Colour: black |
| Salopettes | 100% nylon | Colour: navy and black |
| Softshell jacket | Outside: 100% nylon. Inside: 100% polyester. | Colour: orange and black |
| Primaloft jacket | Outer: 100% nylon Liner: 100% nylon. Filling: 100% polyester fibre. | Colour: orange and black |
| ECW jacket | Outer shell: 85% polyester, 15% cotton. Lining: 100% nylon. Filling: 80% duck down, 20% small feather. Fur trim: 90% acrylic, 10% polyester. | |
| Carhartt jacket | Shell: 100% cotton. Lining: 100% nylon. Interlining: 100% polyester. | Colour: navy |
| Windproof cap | Shell: 100% nylon. Lining: 100% polyester. | Colour: navy and grey |
| Beanie | 100% polyester | Colour: black and orange |
| Balaclava | 100% polyester | Colour: black and orange |
| Neck gaiter | 100% polyester | Colour: grey |
| Goggles | Foam lined, plastic body | |
| ECW mitts | Outer: 94% nylon, 6% spandex. Outer back: 100% polyester. Palm: 100% leather. Shell liner: 100% polyester. Removable liner: 68% polyester, 32% olefin. | |
| Kinco gloves | Leather | |



| | |
|---|---|
| Fleece gloves | Shell: 96% polyester, 4% spandex. Palm: 50% nylon, 50% PU |
| Wool gloves/mitts | 100% wool |
| Merino gloves | 100% merino |
| Polypro gloves | Polypropylene |
| ECW boots | Synthetic and leather uppers |
| Sorel boots | Natural rubber and leather |
| Boot liners | 100% synthetic |
| Socks | 100% wool |

Table A1: Catalogued material used in New Zealand Antarctic program gear.



| Site | Morphotype | Colour | Size (μm) | Polymer |
|------|-----------|--------|-----------|---------|
| S1 | fibre | blue | 630 | PET |
| | fragment | blue | 226 | acrylic copolymer |
| | fragment | pink | 326 | acrylic melamine copolymer |
| | fibre | blue | 318 | acrylic epoxy resin copolymer |
| S2 | fibre | blue | 541 | polyamide |
| S3 | fibre | blue | 1330 | polyamide |
| | fibre | blue | 885 | PET |
| | fibre | clear | 256 | PMMA |
| | fibre | blue | 1034 | PET |
| | fibre | blue | 500 | PET |
| | fibre | blue | 700 | PET |
| | fibre | red | 139 | PA |
| | fibre | clear | 1519 | PMMA |
| S4 | fibre | blue | 273 | polyvinyl acetate copolymer |
| | fibre | clear | 457 | PET |
| | fragment | blue | 166 | PE |
| | fragment | blue | 168 | silicone |
| | fragment | pink | 130 | alkyds |
| | fragment | blue | 90 | polyethylene |
| | fragment | pink | 221 | polyethylene |
| S5 | fibre | blue | 1224 | PET |
| | fibre | blue | 883 | PET |
| | fibre | blue | 419 | PET |
| | fibre | blue | 329 | PET |
| | fragment | pink | 313 | PET |
| | fragment | pink | 144 | PMMA |
| S6 | fragment | pink | 405 | polypropylene |
| | fibre | pink | 252 | PMMA |
| | fibre | blue | 255 | polyamide |
| | fibre | clear | 544 | PMMA |



| | | | | |
|---|---|---|---|---|
| S7 | fibre | blue | 519 | polyvinylidene |
| | fragment | pink | 225 | PET |
| | fibre | blue | 751 | PET |
| | fragment | pink | 189 | polyethylene |
| | fibre | clear | 550 | polyester copolymer |
| | fibre | blue | 905 | PET |
| | fibre | pink | 3510 | polyamide |
| | fibre | clear | 1534 | PET |
| | fragment | blue | 89 | cellulose nitrate |
| | fragment | pink | 215 | acrylate copolymer |
| S8 | fibre | blue | 2453 | PET |
| | fragment | blue | 744 | acrylic copolymer |
| | fragment | blue | 981 | acrylic copolymer |
| | fibre | blue | 660 | acrylic copolymer |
| | fibre | pink | 258 | acrylic copolymer |
| S9 | fragment | white | 519 | cellulose nitrate |
| S10 | fibre | pink | 185 | polyamide |
| | fragment | blue | 77 | PMMA |
| | fragment | blue | 63 | PMMA |
| S11 | fibre | clear | 396 | PET |
| | fragment | purple | 85 | PET |
| S12 | fibre | clear | 889 | PET |
| | fibre | clear | 498 | PTFE |
| S13 | fibre | blue | 617 | polyamide copolymer |
| | fibre | blue | 1388 | cellulose nitrate |
| | fibre | blue | 730 | polyamide |
| | fragment | pink | 314 | PET |
| | fibre | pink | 468 | PET |
| | fragment | pink | 258 | acrylic copolymer |
| S14 | film | blue | 1497 | PET |
| | fragment | green | 199 | alkyds |




| | fibre | blue | 1092 | PET |
|---|---|---|---|---|
| | fibre | blue | 1453 | PET |
| | fibre | blue | 1647 | alkyds |
| | fragment | pink | 135 | PMMA |
| | fragment | blue | 142 | acrylic copolymer |
| | fibre | pink | 2003 | PET |
| | fibre | blue | 1131 | cellulose nitrate |
| | fibre | clear | 1841 | PET |
| S15 | fibre | clear | 960 | PET |
| | fibre | blue | 2005 | PMMA |
| | fibre | pink | 983 | PET |
| S16 | fibre | pink | 726 | PET |
| | fibre | blue | 1322 | methyl vinyl ether |
| | fibre | pink | 415 | PET |
| | fragment | blue | 208 | PVC |
| | fragment | green | 180 | alkyds |
| | fibre | blue | 722 | PET |
| | fibre | blue | 200 | styrene copolymer |
| | fibre | clear | 300 | PET |
| | fibre | clear | 1379 | PET |
| | fibre | clear | 1462 | PET |
| | fibre | blue | 421 | poly(methacrylic anhydride) |
| | fragment | green | 346 | acrylic copolymer |
| S17 | fibre | blue | 400 | PET |
| | fragment | blue | 50 | PET |
| | fragment | blue | 68 | PMMA |
| | fragment | grey | 336 | PTFE |
| | fragment | grey | 63 | PFTE |
| | fragment | green | 121 | acrylic copolymer |
| S18 | fragment | blue | 110 | PVC |
| | fibre | blue | 1458 | PET |
| | fibre | clear | 484 | PET |
| | fibre | blue and white | 1327 | PET |





|  | | | | |
|---|---|---|---|---|
|  | fragment | blue | 123 | PVC |
|  | fibre | pink | 467 | PET |
|  | fibre | blue | 465 | PET |
|  | fibre | blue | 404 | PET |
|  | fragment | blue | 128 | PVC |
| S19 | fibre | blue | 462 | PET |
|  | fibre | blue | 901 | PVC |
|  | fibre | pink | 283 | PET |
|  | fragment | pink | 171 | PET |
|  | fragment | pink | 118 | PET |
|  | fragment | blue | 164 | acrylic copolymer |
|  | fragment | blue | 113 | PVC |
|  | fragment | blue | 116 | PVC |
|  | fragment | blue | 121 | PVC |
|  | fragment | blue | 165 | PVC |

Table A2: Characteristics of microplastics identified. As discussed in the text, 'blue' includes blue, black and navy. 'Size' indicates the width for fragments and length for fibres.
| Site number | 6-hour distance (km) | 24-hour distance (km) | 156-hour distance (km) |
|---|---|---|---|
| S1 | 202 | 849 | 5737 |
| S2 | 173 | 752 | 5972 |
| S3 | 77 | 328 | 5006 |
| S4 | 77 | 328 | 4985 |
| S5 | 87 | 302 | 4660 |
| S6 | 305 | 681 | 4004 |
| S7 | 346 | 1003 | 4172 |
| S8 | 285 | 1010 | 4019 |
| S9 | 281 | 1060 | 4243 |
| S10 | 285 | 1009 | 4127 |
| S11 | 305 | 1243 | 5316 |
| S12 | 43 | 321 | 4540 |
| S13 | 45 | 321 | 4520 |
| S14 | 52 | 323 | 4780 |
| S15 | 60 | 320 | 4723 |
| S16 | 65 | 315 | 5026 |
| S17 | 65 | 315 | 4777 |
| S18 | 68 | 304 | 4892 |
| S19 | 68 | 304 | 4882 |

**Table A3.** Mean distance travelled by air masses for trajectory ensembles at each of the sites, at 6 hours, 24 hours and 156 hours prior to sampling. Trajectories were initialised at a starting height of 2000 m; starting heights of 1000 m and 500 m yielded similar results.

*Author contributions.* Conceptualization: LER and SG. Methodology: LER, HR and SG. Field sampling: ARA, ML, AJM. Investigation: ARA, NEW, AS. Visualization: ARA, NEW, ML, AS. Supervision: LER, SG. Writing – original draft: ARA, LER. Writing – review and editing: ARA, LER, SG, HR, AS, NEW, ML, AJM.

*Competing interests.* The authors declare no competing interests.

*Acknowledgements.* The authors thank Nicole Lauren-Manuera, Alex Nicholls, Paula Brooksby, Justin Harrison, Antarctica New Zealand, staff and students from Gateway Antarctica and the Postgraduate Certificate of Antarctic Studies 2019 group. LER and SG were supported by the Royal Society of New Zealand Marsden Fund (contract number MFP-UOC1903). ARA was supported by Gateway Antarctica's Ministry of Foreign Affairs and Trade Scholarship in Antarctic and Southern Ocean Studies. We acknowledge mana whenua, Ngāi Tūāhuriri, on
whose lands our analysis and writing took place.





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
