# Peer review of "Fig. S1. Microplastic concentrations identified at each sample site from the Ross Island region of Antarctica"

_The Cryosphere, 2021_

## Author Comment (AC1)

*We thank the reviewer for their helpful and constructive comments. The reviewers comments are shown in black, our response in blue italics, and amendments to the text are in red.*

Response to reviewer 1:

So far as I know (and I am not an expert) this is the first documentation of microplastics in Antarctic snow. As such it is a valuable paper that documents the ever growing reach of this pollutant.

Analysis of air borne microplastics is a relatively new field and one where protocols are still being developed. I am pleased to see that considerable thought has gone into the minimising of contamination in this study. The sampling and analysis protocols are thoroughly described and rigorous, providing confidence that the results of microplastic concentrations are accurate. The discussion of the potential sources is thoughtful and realistic. I have some specific comments below that might be considered before final publication about the analysis and the sources.

The analysis method (section 2.3) involves visual identification of the microplastics followed by FTIR characterisation. This visual approach must necessarily preclude some very small microplastic fragments, but there is no discussion of a lower size limit. The useful effort to check recovery focusses on particles of 500µm. In Figure 5 the lowest size range is 0-200 µm. Around line 240 there is discussion of the possible bias against detecting small particles in this work, but I would suggest that this be discussed in the methods section.

*To address this, two sentences have been added to line 80 and 88, respectively:*

"It is recognised that due to human error, inability to transfer some particles due to the small size and brittleness, and the translucent and transparent nature of some microplastics that there are limitations to this method which are unavoidable. This may lead to the underestimation of microplastics in this study."

"The smallest particle identified in this study was 44 µm (non-plastic), meaning particles less than this size were not accounted for due to the workable limit."

Table A2 describes the size of particles, but particularly for fibres with one long and another short axis, the issue of size is ambiguous and the caption of this table could be expanded to clarify this.

*This may have been a misunderstanding; we state in the caption for Table A2 that: "'Size' indicates the width for fragments and length for fibres."*

The discussion of sources is thoughtful and interesting, although necessarily inconclusive. As I understand it the remote sampling sites are generally south and west of the main nearby stations (Mcmurdo and Scott   Figure 3) and the airflow was generally from the south and east (Figure 6). In line 357 I think the authors imply that air masses containing the sampled snow would have passed over "the bases" before reaching the deposition sites and in line 300 they suggest the bases are the main source. Their data shows higher microplastics closer to the bases, so there clearly is a source there, but I'm not sure that their data does imply the bases are the sources for the microplastics at the sampling sites further from the Scott and McMurdo stations. I would say you cannot conclude if the source is from there or from very long range transport, but maybe I am missing something in the argument.

*We have reworded this sentence to address this issue on line 300 and line 357 respectively:*

"Short-range transport of microplastics from the bases to  sampling sites close by (e.g. S14-S19) is more likely than long-range transport, given the sites' proximity to research bases and the climatology of the area. Yet sites further away may have more influence from long-range transportation, showing the potential influence of both short range and long range inputs."

"…it is likely that local inputs were a contributor to the microplastic pollution identified. "

We do know, as the authors document, that long range transport of other material than microplastics does occur to Antarctica, so this is clearly a potential source. In that context I did not really understand in line 203 what the authors mean by the residence time. I think their figure of 156 hours is the longest trajectory they considered. However, assuming that microplastics can remain suspended in the air (my understanding of the term residence time) for longer than that, they could have been derived from further afield, or indeed have been deposited and resuspended from land or the sea en route. I would suggest the argument here might be clarified.

*These values have been taken from previously hypothesised residence times of airborne microplastics, as shown in the following paper: Brahney, J., Mahowald, N., Prank, M., Cornwell, G., Klimont, Z., Matsui, H. and Prather, K.A., 2021. Constraining the atmospheric limb of the plastic cycle. Proceedings of the National Academy of Sciences, 118(16).*

*At the first mention of residence time in line 182 we have added the following to expand on and clarify this:*

"The residence time is the length of time that a particle can remain in the atmosphere which was estimated by Brahney et al. (2021) to be as long as 156 hours prior to deposition at the sampling site. We acknowledge that microplastics could be suspended in air for longer than the time periods used, although highly unlikely, and that they could have been derived from further afield or have been deposited and resuspended from land or the sea enroute."

References

Brahney J, Mahowald N, Prank M, Cornwell G, Klimont Z, Matsui H, et al. Constraining the atmospheric limb of the plastic cycle. Proceedings of the National Academy of Sciences 2021; 118.

---

## Author Comment (AC2)

*We thank the reviewer for their helpful and constructive comments, which we have addressed as described below. Their comments are shown in black, our response in blue italics, and amendments to the text are in red.*

Response to Reviewer 2:

Overall this is a well written manuscript. However, there are some places where the results and methods can be more clearly presented. Additionally, the authors should more clearly state the limitations of their approach. Lastly, I cannot find any data availability statement. I recommend major revisions.

One aspect of the methods that was not clear to me is the calculation of microplastic count per liter. Was this from the total liquid volume? If so, the authors must provide that information and this should be stated in the methods (line 60). Additionally, I suggest the authors add a summary table summarizing how many plastic particles per sample were identified and the volume of water in the main text.

*Yes, this was from the total liquid volume. This has now been added and a column added to table A3 with sample volume (please see addition to table at the end of document):*

"Snow samples were thawed at room temperature for 24–48 hours prior to analysis. Thawed samples were filtered through a glass apparatus attached to a vacuum using a cellulose nitrate membrane filter (Whatman nitrocellulose membrane, 50 mm diameter, 0.45 µm pore size). The volume of liquid (melted snow) was recorded at this step, before rinsing, to establish the volume of each individual sample bottle. Approximately 10–20 mL of 70% ethanol was used to rinse the filters, and a further 10–30 mL of 96% ethanol was added to soak the filter for 10 minutes, to prevent bacterial and viral growth for further biosecurity measures. The sides of the glassware were thoroughly rinsed with ultra-pure water (<18 MΩ) to dislodge any microplastics adhered to the walls of the filtering equipment, and samples were dried under vacuum."

Blank corrections are really important here. On average, it appears that 6 particles were identified in the blanks. This seems to be greater than number of plastic particles measured in some samples (E.g. S2, S9). This should be clearly stated, and this highlights the importance of the approach used for blank corrections. There are many different approaches in the microplastics literature for handling blanks and for handling low sample counts (e.g., Bender et al., 2020 Applied Spectroscopy; Miller et al., 2021 Journal of Hazardous Material; Standard Operating Procedures for Extraction and Measurement by Infrared Spectroscopy of Microplastic Particles in Drinking Water by the California Water Board) including the method used here and using FTIR spectral matches. I suggest the authors include a citation for how they chose their approach. Additionally, the authors should provide some additional information such as how many particles from field and laboratory procedures were identified per sample and what the blank particles look like (color, size, morphology, spectra, etc).

*Thank you for your comment. We followed the methodology highlighted in Brander et al. (2020) and used in Vandermeersch et al. (2015), whereby the number of particles matching the identical characteristics of those found in the blanks (field blank and laboratory blank samples) were omitted from further analysis and the total number of particles found in corresponding daily blank samples (with identical characteristics) were subtracted off the final results, to ensure a conservative approach. All fragments identified in field and laboratory blanks matched the colour and coating of the sampling bottles and were disregarded throughout the analysis. No fragments were found in the daily blanks, meaning fibres were the predominant morphotype.*

*We have added the following to line 106 and made the following changes to section 3.1, respectively, to highlight the approach taken for blank corrections:*

*Line 106:* "For blank corrections we followed the methodology highlighted in Brander et al. (2020) and used in Vandermeersch et al. (2015), whereby particles matching the identical characteristics of those found in the blanks (laboratory and field blanks) were omitted from further analysis, and total daily laboratory blank findings were subtracted from the corresponding samples (Table. S1)."

*Section 3.1:* "Recovery rates for spiked samples were 100%. Across the sample controls an average of 1.5 (±0.89) particles were found in daily laboratory blanks (*n*=11), 3 (±0) particles in field blanks (*n*=2) and 2 (±1) particles in method controls (*n*=2). *All fragments identified in field and laboratory blanks matched the colour and coating of the sampling bottles. No fragments were found in the daily blanks, meaning fibres were the predominant morphotype (Table. S1).* Spectroscopic analysis confirmed that the outside bottle coating and fragments found in the blanks were polymethyl methacrylate (PMMA). Suspected microplastic particles in field samples of an identical colour and morphotype to those detected in the field and laboratory blanks were not analyzed further and discounted from the results. All reported PMMA still included in results was a different morphotype and colour to the corresponding sampling bottles, and therefore some PMMA is still shown in results. Particles with identical characteristics of those found in the field and laboratory blanks were excluded from further study and daily blank contamination was subtracted from the results of corresponding samples."

*We have also moved the following lines from section 2.4.3 to the previous section, 2.4.2 so all information is in one place:*

"Daily laboratory blanks were analyzed using the same procedure as all samples with 500 mL of ultra-pure water."

"The laboratory blanks were analyzed to compare results for each individual date and accounted for in the data for the samples filtered on the corresponding days."

"Particles found in field samples with identical characteristics to those found in blanks were discarded and excluded from the results."

*Regarding the interpretation that there were 6 particles on average in the blanks. There were on average 1 in the daily blanks, 2 in the method controls and 3 in the field blanks, which we have chosen not to combine as they are all addressing contamination gained at different stages of the process, i.e. the method controls and field blanks were likely to also pick up contamination from the laboratory process, just as the daily blank did. The daily blanks corresponded with certain samples that were processed on different days, so these are unique to those samples (see added table at end of document). The standard deviations have also been added to the three values provided (see additions to section 3.1 above).*

Additionally line 140 to 141 should include standard deviations of the blanks and the authors should include a table like A2 for the particles identified in the blanks which will also help the reader to understand lines 144 to 145. It appears that sample volume is a limitation here, as a greater sample volume would have resulted in particle counts that were greater than the blank values. This should be clearly stated and perhaps samples with low microplastic counts should be clearly identified. I suggest the authors clearly state this limitation of the data set in the text and in the conclusion and make recommendations for future studies to collect a greater sample volume. For example, I suggest lines 213 to 215 should state "our work provides the first evidence of microplastics in Antarctic snow. limitations of this dataset

include low sample volume and therefore should be replicated, however, our preliminary results suggest…" This low sample volume also explain why particles are higher than prior work.

*The standard deviations have been added to line 140 and 141, as shown above to edits made in section 3.1.*

*A table has been added to the supporting information with the blank particle characteristics, this can be found at the end of the document.*

*We agree with the reviewer that there are limitations experienced due to the low sample volume. This low sample volume was due to the import process of biological samples necessary from the Antarctic into New Zealand, with permits restricting the amount of liquid we were able to bring back. In response to this, we have added in further explanation of this:*

*Line 221:* "This dataset is limited by the low sample volumes due to the permitting restrictions for Antarctic samples. Therefore, we recommend this study to be replicated to further understand these preliminary findings. Larger volumes of snow (≥10 L) or replicates from the same study sites would be beneficial for future research."

While picking putative plastic particles is a good approach, it is important that the authors note that there is a limitation to this method. Specifically that it is really hard to detect translucent or transparent microplastics, and that it is really hard to pick small particles (which the authors noted), and many particles become brittle and difficult to transfer. I think it's important to note these limitation, specifically in the discussion about the color.

*To address this we have added in the following to section 2.3, Line 80:*

"It is recognised that due to human error, inability to transfer some particles due to the small size and brittleness, and the translucent and transparent nature of some microplastics that there are limitations to this method which are hard to avoid. This may lead to the underestimation of microplastics in this study."

Lastly, there is no data availability statement. Please see Cowger et al., 2020 Critical Review of Processing and Classification Techniques for Images and Spectra in Microplastic Research, Applied spectroscopy for a discussion on data sharing practices for microplastic data.

*A data availability statement has now been added at the end of the paper after line 370:*

"The microplastic data generated in this study has been provided in the appendix of this manuscript, including microplastic counts, sample volume, particle size, shape and polymer type. Relevant data to evaluate the conclusions of this paper are present in either the main paper, the appendix, or the supporting information provided."

Minor comments:

Line 30: There is evidence that anthropogenic pollutants in ice core records from 1889 (e.g. McConnell et al., 2014, Scientific reports)

*We have amended this statement on line 30:*

"With a few exceptions, such as lead pollution in the late 19th century (McConnell et al., 2014), Antarctica was generally thought to be largely untouched by humans until the early 20th century due to its inaccessibility, extreme environmental conditions and barriers such as the Antarctic Circumpolar Current (Tin et al., 2014; Gordon, 1971). While the human footprint has increased over the last century, Antarctica is still a place of peace and science and is thought of as the last remaining true wilderness on earth (Tin et al., 2016). Due to this, Antarctica can act as an indicator of physical, chemical, and biological effects caused from anthropogenic stresses (Huiskes et al., 2006)."

Line 52: was the funnel also stainless steel? And is there an approximate area of the surface snow that was sampled?

*Yes, the funnel was stainless steel. We have added this to line 52. The area of surface snow was not measured so we have chosen not to include this.*

"Samples were collected using a stainless-steel scoop and stainless-steel funnel to fill each 500 mL bottle with snow from the top 2 cm of the surface."

Section 2.2: Were the reagents used pre-filtered?

*It was established that adding a step of filtering for reagents could increase the contamination levels by adding another step for sample exposure, so reagents were not pre-filtered, but all reagents used were analytical grade from previously unopened containers. Below statement added in at line 73:*

"All reagents used were previously unopened and analytical grade, with blanks also undergoing identical analysis to control for contamination."

Line 56: were they kept covered during thawing?

*Yes, added in line 56:*

"Snow samples were thawed in the sealed sample bottles at room temperature for 24–48 hours prior to analysis."

Line 68: Was the magnetic stir bar coated in plastic?

*Yes, this was accounted for, added text in line 68:*

"The solution was then left, with the beaker opening covered in aluminium foil, for 20 minutes before being stirred on a hot plate at 45 ◦C for 2–3 hours with a magnetic stir bar. The magnetic stir bar had a white PTFE coating, which was not identified in any field or blank samples."

Line 70: Glass Fiber?

*Added into line 70:*

"…onto a Whatman glass fibre GF/C filter…"

Line 76: I would rewrite to say "Suspected microplastics were characterized…"

*This has been edited at line 76:*

"Suspected microplastics were characterized into four main morphotypes"

Line 85: What is the minimum size that the authors were able to pick?

*Added to line 88:*

"The smallest particle identified in this study was 44 µm (non-plastic), meaning particles less than this size were not accounted for due to analysis limitations."

Line 88 to 89: This is unclear to the reader. I suggest defining the acronyms.

*These have been added:*

(Databases: Industrial chemicals, pure organic compounds; organosilicons; polymers, Hummel defined basic; Sadtler acrylates & Methacrylates; Sadtler fibers & textile chemicals; Sadtler fibers by microscope; Sadtler inorganics; Sadtler polymers & monomers (comprehensive); saddler polymers, Hummel; Sadtler standards (organic & polymeric compounds subset); Sigma-Aldrich library of FT-IR spectra).

Line 90 to 91: Was there any smoothing, baseline correction, atmospheric suppression, etc conducted on the spectra?

*Baseline correction was applied, and $CO_2$ spectral ranges excluded, this has been added line 87:*

"All spectra were baseline corrected and $CO_2$ spectral ranged excluded, saved, and compared against the following Wiley spectral libraries"

Line 90 to 91: For the spectra that did not match the library, can the authors provide some additional detail about the matching approach? Perhaps an example spectra and subsequent match would be helpful here.

*We have added in the following reference to explain the necessity of manual peak picking in environmental plastic studies when using spectral libraries:*

"Those <70% that exhibited plastic characteristics from visual screening and possessed similar µFTIR spectra were analyzed further by the authors using peak picking tools to identify characteristic peaks of plastic polymer types (Kroon et al., 2018). Due to the environmental degradation sampled particles have been subjected to, and the limitations of spectral libraries due to the use of high standard polymers, visual inspection of spectra is an essential step in identification of environmental microplastic analysis (De Frond et al., 2021; Shim et al., 2017)."

Line 96 to 97: What was the lid of the sample bottle made of?

*This was clear silicone. Have added this information into line 143:*

"The lid of the sampling bottles was confirmed as clear silicone, which was not found in any of the samples or blanks."

Line 156 to 157: Does this include the plastics with matches <70%?

*Yes. This has been highlighted in section 2.3 and the following added into line 91 (also refer to above additions for lines 91):*

"These particles matching characteristic spectral peaks were included in results."

Figure 4: I think the abbreviations used in the figure should be defined in the figure caption.

*These definitions have been added into the caption of Figure 4.*

"Figure 4. Polymer types identified across all samples. (a) Number of each polymer type found across all Antarctic study sites (PET: polyethylene terephthalate, CP: copolymers, PMMA: polymethyl methacrylate, PVC: polyvinyl chloride, PA: polyamide, other: (polytetrafluoroethylene (PTFE), polyvinylidene, polypropylene, silicone and polymethyl anhydride.), PE: polyethylene, ALK: alkyds, CN: cellulose nitrate);. (b) Number of microplastic fragments and fibres identified in each colour category (films ($n$=1) excluded);. (c) Size distribution of microplastics across all samples categorized by morphotype (length)."

Figure 5: I suggest combine with figure 4.

*This has now been combined in the updated manuscript, with figure heading updated as shown in above comment.*

Line 255 to 256: I suggest reminding the reader the minimum and max distances to these stations.

*The following has been added on line 256:*

"Antarctic research stations on Ross Island, Scott Base (NZ) and McMurdo Station (US), have the closest proximity to the sampling sites, up to 20 km away (Fig. 1), with Zucchelli Station (Italian) the next closest at 350 km away."

Section 4.3.1: Are tumble dryers used at the stations? If so, perhaps considering them as a potential source (see Tao et al., 2022, Microfibers Released into the Air from a Household Tumble Dryer, ES&T).

*Yes, the following has been added to line 281:*

"An excess of 700,000 synthetic fibres are released from an average 6 kg load of washing acrylic fibres (Napper and Thompson, 2016) and with tumble dryers being present these may also contribute to the presence of microplastics (Tao et al., 2022). Future work should focus on quantifying the contribution of tumble dryers and waste-water discharge to the abundance of microplastics in Antarctica."

Line 281 to 282: what are the WWT processes for the bases nearby the sampling sites?

*We do not have enough information to make claims on this, but we have added the following to line 283:*
"Future work should focus on the most effective wastewater treatment process(es) for microplastic removal, which could ultimately be used at these bases."

Table A2: Characteristics of microplastics identified. Volume is the melted snow content of each sample. As discussed in the text, 'blue' includes blue, black and navy. 'Size' indicates the width for fragments and length for fibres.

| Sampling Site | Morphotype | Colour | Size | Polymer | Volume (mL) |
|---|---|---|---|---|---|
| | | | | | |
| S1 | fibre | blue | 630 | Polyester | 207 |
| | fragment | blue | 226.24 | acrylic copolymer | |
| | fragment | pink | 326.34 | acrylic melamine copolymer | |
| | fibre | blue | 318.31 | acrylic epoxy resin copolymer | |
| | | | | | |
| S2 | fibre | blue | 540.86 | Polyamide | 250 |
| | | | | | |
| S3 | fibre | blue | 1329.7 | Polyamide | 233 |
| | fibre | blue | 885.18 | Polyester | |
| | fibre | clear | 255.99 | PMMA | |
| | fibre | blue | 1033.72 | Polyester | |
| | fibre | blue | 499.85 | Polyester | |
| | fibre | blue | 699.56 | Polyester | |
| | fibre | red | 138.84 | PA | |
| | fibre | clear | 1519.3 | PMMA | |
| | | | | | |
| S4 | fibre | blue | 272.57 | polyvinyl acetate copolymer | 192 |
| | fibre | clear | 457.24 | Polyester | |
| | fragment | blue | 166.24 | PE | |
| | fragment | blue | 167.65 | silicone | |
| | fragment | pink | 129.98 | alkyds | |
| | fragment | blue | 89.77 | polyethylene | |
| | fragment | pink | 220.84 | polyethylene | |
| | | | | | |
| S5 | fibre | blue | 1224.39 | Polyester | 216 |
| | fibre | blue | 883.31 | Polyester | |
| | fibre | blue | 418.53 | Polyester | |
| | fibre | blue | 329.37 | Polyester | |
| | fragment | pink | 313.34 | Polyester | |
| | fragment | pink | 144.48 | PMMA | |
| | | | | | |
| S6 | fragment | pink | 415.38 | polypropylene | 208 |
| | fibre | pink | 252.04 | PMMA | |
| | fibre | blue | 254.84 | polyamide | |

| | | | | | |
|---|---|---|---|---|---|
| | fibre | clear | 544.04 | PMMA | |
| | | | | | |
| **S7** | fibre | blue | 518.56 | polyvinylidene | 184 |
| | fragment | pink | 224.51 | Polyester | |
| | fibre | blue | 751.09 | Polyester | |
| | fragment | pink | 188.61 | polyethylene | |
| | fibre | clear | 550.18 | polyester copolymer | |
| | fibre | blue | 904.05 | Polyester | |
| | fibre | pink | 3510.5 | polyamide | |
| | fibre | clear | 1533.8 | Polyester | |
| | fragment | blue | 89.02 | cellulose nitrate | |
| | fragment | pink | 215.11 | acrylate copolymer | |
| | | | | | |
| **S8** | fibre | blue | 2452.53 | Polyester | 221 |
| | fragment | blue | 744.33 | acrylic copolymer | |
| | fragment | blue | 981.47 | acrylic copolymer | |
| | fibre | blue | 660.42 | acrylic copolymer | |
| | fibre | pink | 257.98 | acrylic copolymer | |
| | | | | | |
| **S9** | Fragment | white | 518.89 | cellulose nitrate | 203 |
| | | | | | |
| **S10** | fibre | pink | 184.84 | polyamide | 191 |
| | fragment | blue | 77 | PMMA | |
| | fragment | blue | 63 | PMMA | |
| | | | | | |
| **S11** | Fibre | clear | 395.78 | Polyester | 235 |
| | fragment | purple | 85.24 | Polyester | |
| | | | | | |
| **S12** | fibre | clear | 888.54 | Polyester | 226 |
| | fibre | clear | 497.85 | PTFE | |
| | | | | | |
| **S13** | Fibre | Blue | 616.95 | polyamide copolymer | 166 |
| | Fibre | Blue | 1387.64 | cellulose nitrate | |
| | Fibre | Blue | 729.62 | polyamide | |
| | Fragment | pink | 314.09 | Polyester | |
| | fibre | pink | 467.88 | Polyester | |
| | fragment | pink | 257.53 | acrylic copolymer | |
| | | | | | |
| **S14** | Film | Blue | 1497.11 | Polyester | 219 |
| | Fragment | Green | 199.48 | alkyds | |
| | Fibre | Blue | 1091.78 | Polyester | |
| | fibre | blue | 1452.55 | Polyester | |
| | fibre | blue | 1647.14 | alkyds | |

| | | | | | |
|---|---|---|---|---|---|
| | fragment | pink | 135.34 | PMMA | |
| | fragment | blue | 142.3 | acrylic copolymer | |
| | fibre | pink | 2002.91 | Polyester | |
| | fibre | blue | 1131.31 | cellulose nitrate | |
| | fibre | clear | 1851.13 | Polyester | |
| | | | | | |
| S15 | Fibre | clear | 959.67 | Polyester | 217 |
| | Fibre | blue | 2005.45 | PMMA | |
| | Fibre | pink | 983.33 | Polyester | |
| | | | | | |
| S16 | fibre | pink | 726.42 | Polyester | 146 |
| | fibre | blue | 1321.59 | methyl vinyl ether/maleic anhydride copolymer | |
| | fibre | pink | 415.09 | Polyester | |
| | fragment | blue | 208.09 | PVC | |
| | fragment | green | 179.78 | alkyds | |
| | fibre | blue | 721.63 | Polyester | |
| | fibre | blue | 200 | Styrene copolymer | |
| | Fibre | Clear | 300 | Polyester | |
| | Fibre | Clear | 1378.89 | Polyester | |
| | Fibre | Clear | 1461.7 | Polyester | |
| | Fibre | Blue | 420.63 | poly(methacrylic anhydride) | |
| | Fragment | Bright green | 346.03 | acrylic copolymer | |
| | | | | | |
| S17 | fibre | blue | 400.2 | Polyester | 156 |
| | fragment | bue | 50.05 | Polyester | |
| | fragment | blue | 67.68 | PMMA | |
| | fragment | grey | 336.11 | PTFE | |
| | fragment | grey | 63.31 | PTFE | |
| | fragment | green | 120.75 | acrylic copolymer | |
| | | | | | |
| S18 | Fragment | Blue | 109.57 | PVC | 160 |
| | Fibre | Blue | 1457.75 | Polyester | |
| | Fibre | Clear | 483.56 | Polyester | |
| | Fibre | Blue and White | 1327.49 | Polyester | |
| | Fragment | Blue | 123.2 | PVC | |
| | Fibre | Pink | 466.68 | Polyester | |
| | Fibre | Blue | 464.9 | Polyester | |
| | Fibre | Blue | 403.55 | Polyester | |

| | Fragment | Blue | 127.71 | PVC | |
|---|---|---|---|---|---|
| | | | | | |
| **S19** | Fibre | Blue | 461.68 | Polyester | 182 |
| | Fibre | Blue | 900.61 | PVC | |
| | Fibre | Pink | 238.21 | Polyester | |
| | Fragment | Pink | 170.81 | Polyester | |
| | Fragment | Pink | 117.6 | PVC | |
| | Fragment | Blue | 163.57 | acrylic copolymer | |
| | Fragment | Blue | 113.41 | PVC | |
| | Fragment | Blue | 116.39 | PVC | |
| | Fibre | Blue | 120.96 | PVC | |
| | Fragment | Blue | 165.47 | PVC | |

*Supporting Information*

*Table S1.* Blank findings from field blanks (FB), laboratory blanks (LB) and daily blanks (DB). Daily blank samples correspond to specific samples processed at the same time and therefore account for contamination of specific samples. Blank corrections were made by subtracting the corresponding daily blank findings from the results before reporting. Fragments were excluded from this table as they were only found in FB and LB, with all matching the colour coating of the sampling bottles.

| Blanks (+ corresponding samples) | Morphotype | Colour | Total |
|---|---|---|---|
| FB1 | Fibre | Black
Clear
Clear | 3 |
| FB2 | Fibre | Black
Blue
Blue | 3 |
| LB1 | Fibre | Blue | 1 |
| LB2 | Fibre | Blue
Clear
Clear | 3 |
| DB1 (S1,S8) | Fibre | Clear
Clear | 2 |
| DB2 (S2,S3) | Fibre | Clear
Clear | 2 |
| DB3 (S6,S7) | Fibre | Blue
Clear
Clear | 3 |
| DB4 (S4, S15) | Fibre | Clear
Clear | 2 |
| DB5 (S9, S11, S12) | Fibre | Blue
Clear | 2 |

| DB6 (S5,S10) | Fibre | Black Clear | 2 |
|---|---|---|---|
| DB7 (S17) | None | - | 0 |
| DB8 (S13) | Fibre | Blue | 1 |
| DB9 (S14) | Fibre | Blue | 1 |
| DB10 (S16, S18) | None | - | 0 |
| DB11 (S19) | Fibre | Blue Clear | 2 |

**References**

Brander SM, Renick VC, Foley MM, Steele C, Woo M, Lusher A, et al. Sampling and quality assurance and quality control: a guide for scientists investigating the occurrence of microplastics across matrices. Applied Spectroscopy 2020; 74: 1099-1125.

De Frond H, Rubinovitz R, Rochman CM. µATR-FTIR Spectral Libraries of Plastic Particles (FLOPP and FLOPP-e) for the Analysis of Microplastics. Analytical Chemistry 2021; 93: 15878-15885.

Kroon F, Motti C, Talbot S, Sobral P, Puotinen M. A workflow for improving estimates of microplastic contamination in marine waters: A case study from North-Western Australia. Environmental Pollution 2018; 238: 26-38.

McConnell JR, Maselli OJ, Sigl M, Vallelonga P, Neumann T, Anschütz H, et al. Antarctic-wide array of high-resolution ice core records reveals pervasive lead pollution began in 1889 and persists today. Scientific Reports 2014; 4: 1-5.

Shim WJ, Hong SH, Eo SE. Identification methods in microplastic analysis: a review. Analytical methods 2017; 9: 1384-1391.

Tao D, Zhang K, Xu S, Lin H, Liu Y, Kang J, et al. Microfibers Released into the Air from a Household Tumble Dryer. Environmental Science & Technology Letters 2022.

Vandermeersch G, Van Cauwenberghe L, Janssen CR, Marques A, Granby K, Fait G, et al. A critical view on microplastic quantification in aquatic organisms. Environmental Research 2015; 143: 46-55.